

# A four-year cardiovascular risk score for type 2 diabetic inpatients

Dolores Ramírez-Prado[1,2], Antonio Palazón-Bru[1,2],
David Manuel Folgado-de la Rosa[2], María Ángeles Carbonell-Torregrosa[3], Ana María Martínez-Díaz[3], Damian Robert James Martínez-St. John[2] and Vicente Francisco Gil-Guillén[1,2]

[1] Research Unit, Elda Hospital, Elda, Alicante, Spain
[2] Department of Clinical Medicine, Miguel Hernández University, San Juan de Alicante, Alicante, Spain
[3] Emergencies Unit, Elda Hospital, Elda, Alicante, Spain

## ABSTRACT

As cardiovascular risk tables currently in use were constructed using data from the general population, the cardiovascular risk of patients admitted via the hospital emergency department may be underestimated. Accordingly, we constructed a predictive model for the appearance of cardiovascular diseases in patients with type 2 diabetes admitted via the emergency department. We undertook a four-year follow-up of a cohort of 112 adult patients with type 2 diabetes admitted via the emergency department for any cause except patients admitted with acute myocardial infarction, stroke, cancer, or a palliative status. The sample was selected randomly between 2010 and 2012. The primary outcome was time to cardiovascular disease. Other variables (at baseline) were gender, age, heart failure, renal failure, depression, asthma/chronic obstructive pulmonary disease, hypertension, dyslipidaemia, insulin, smoking, admission for cardiovascular causes, pills per day, walking habit, fasting blood glucose and creatinine. A cardiovascular risk table was constructed based on the score to estimate the likelihood of cardiovascular disease. Risk groups were established and the c-statistic was calculated. Over a mean follow-up of 2.31 years, 39 patients had cardiovascular disease (34.8%, 95% CI [26.0–43.6%]). Predictive factors were gender, age, hypertension, renal failure, insulin, admission due to cardiovascular reasons and walking habit. The c-statistic was 0.734 (standard error: 0.049). After validation, this study will provide a tool for the primary health care services to enable the short-term prediction of cardiovascular disease after hospital discharge in patients with type 2 diabetes admitted via the emergency department.

Corresponding author
Antonio Palazón-Bru,
antonio.pb23@gmail.com

## INTRODUCTION

Cardiovascular diseases (CVD) constitute one of the main causes of death worldwide, and one of the main reasons for admission via the hospital emergency department (ED) (*Fan et al., 2011*; *World Health Organization, 2014*). The most important risk factors for CVD include diabetes mellitus, hypertension, dyslipidaemia, obesity and smoking

(*World Health Organization, 2007*). These factors are all prevalent among patients admitted via the ED (*Cinza Sanjurjo et al., 2006*; *Fan et al., 2011*).

One in every six ED admissions among diabetic patients is related to the diabetes itself, with almost half of these admissions due to glycaemic decompensation. The other main reasons (unrelated to the diabetes) for ED admissions among these patients are lesions and poisonings (*Hinojosa Mena-Bernal et al., 2004*).

We are unaware of any studies in patients with type 2 diabetes admitted via the ED that have analyzed the onset of CVD and constructed a predictive model to indicate which of these patients have a greater likelihood of presenting CVD. Although there exist cardiovascular risk tables constructed with data from the general population, health centres, working persons and volunteers, the results of these tables are not based on the follow-up of patients with specific disorders, such as type 2 diabetes (*Cooney, Dudina & Graham, 2009*). Thus, the cardiovascular risk obtained from these tables might be underestimated, as we must consider that diabetic persons admitted via the ED present important differences (highly heterogenic) with the type of patients used for the construction of these scales and tables. For example, diabetic patients admitted via the ED have, a priori, more disorders. Accordingly, we undertook a study with a four-year follow-up at the Elda Hospital (Spain) to construct a predictive model of CVD. Once validated (by reproducing our results in other populations) and after hospital discharge, this model could be used preventively by the primary health care services with the aim of reducing the cardiovascular mortality and morbidity in patients with type 2 diabetes admitted via the ED.

## MATERIALS & METHODS

### Study population, design and participants, ethical considerations

The study population was formed by diabetic patients admitted via the ED in the Valle de Elda healthcare area (Valencian community), an industrial area with 198,090 inhabitants with a low-to-medium socioeconomic level (*Martínez-Orozco et al., 2015*). The ED of Elda Hospital (a public institution) tends to about 160 general emergency cases daily among the adult population, not including obstetric and gynaecologic cases (*Carbonell Torregrosa et al., 2014*).

The study cohort comprised type 2 diabetic patients admitted for any reason via the ED of Elda Hospital (only hospital in the healthcare area), aged >13 years (patients younger than 13 years are seen by the paediatric services), who were willing to participate. The follow-up was four years. Patients were excluded if they were pregnant or had a personal history of acute myocardial infarction, stroke, or cancer, or were receiving palliative care. A random sample was selected from all patients admitted via the ED between January 2010 and March 2012. The sampling procedure involved random selection of one day every week and recruiting all the diabetic patients who fulfilled the criteria and were admitted on that day.

Patients with type 1 diabetes were not included in this study because they generally have different characteristics to patients with type 2 diabetes; for example, patients with type 1
diabetes are usually younger. Grouping together two non-homogenous groups of patients would produce results that would not really be useful in daily clinical practice. For this reason, most authors usually analyze different outcomes depending on the type of diabetes (*Ramírez-Prado et al., 2015*). Each patient was followed from the recruitment date until he or she had a CVD, whether fatal or not. If no CVD developed, the patient was followed for four years (if still alive), or until the date of last clinical contact (assuming the patient had no CVD by this date).

The study posed no additional risk to the patients and an indirect benefit was expected, as the results might reduce short-term cardiovascular morbidity and mortality in this type of patient. The study was carried out in compliance with the principles of the World Medical Association Declaration of Helsinki and complied with the European Union norms of good clinical practice. The patients were informed verbally about the study and about the information required. The study was approved by the Ethics Committee of the Elda Department of Health (Ref. UI13016).

## Variables and measurements

The main outcome variable was cardiovascular morbidity or mortality during the four-year follow-up. Cardiovascular conditions were considered to be those affecting the heart or blood vessels (cerebrovascular, legs, kidneys or heart) (*Bonny et al., 2008*).

Data collected at admission (baseline) included gender, age (years), personal history of diseases (heart failure, renal failure, depression, asthma/chronic obstructive pulmonary disease (COPD), hypertension and dyslipidaemia), use of insulin, smoking, admission due to cardiovascular reasons, number of tablets per day (usual medication for whatever condition, excluding diabetes therapy), walking habit, fasting blood glucose (mmol/L) and creatinine (μmol/L).

Cardiovascular morbidity and mortality was assessed during the four-year follow-up by regularly checking the hospital and health centre records. In the case of any doubt about death, contact was made with the patient (if alive) or the patient's relatives, or by contacting the patient's assigned physician (if there was still doubt). Information about gender, age, personal history of diseases, smoking, taking of insulin, and number of pills daily was obtained by patient interview and corroborated from the medical records. Information about walking habits was obtained just at the interview. Data regarding admissions were obtained from the hospital records. The baseline fasting blood glucose and the creatinine were measured according to the current clinical guidelines (*American Diabetes Association, 2014*).

## Sample size and statistical methods

The final cohort sample was 112 patients. Assuming 95% confidence, an expected censored proportion of 60%, an exposure proportion of 35% and an expected hazard ratio (HR) of 2.50, the power to contrast a HR different to 1 was calculated. The resulting value, obtained from implementing the formula for the power in an Excel spreadsheet and solving it with the Solver tool, was 83.28%.

As smoking and walking had lost values, 32.4% and 24.1% respectively, 100 multiple imputations were made beforehand using logistic regression switching with predictive

mean matching. This is the most suitable procedure when the number of missing data is between 10–50%. In this way we were able to work with all the variables (*Marshall, Altman & Holder, 2010*).

Absolute and relative frequencies were used to describe the qualitative variables, with means and standard deviations for the quantitative variables. A Cox multivariate regression model was constructed to determine which variables were associated with cardiovascular morbidity and mortality, calculating the HR. As we had few patients, we selected a maximum number of explanatory variables in the model. As a heuristic rule we considered there needed to be at least 10 observations of *morbidity and mortality* or *no morbidity and mortality* for each explanatory variable. To obtain the variables in the model we analyzed all the possible combinations with a maximum of 7 variables (16,383), calculating the value of the c-statistic in all of them. The combination with the highest value was then selected. The c-statistic is similar to the area under the ROC curve, but the former takes into account censoring. The goodness of fit of the model was assessed by the score (log-rank) test. Using the $\beta$ coefficients of the multivariate model a risk table was constructed based on the sum of the points to estimate the likelihood of CVD (*Sullivan, Massaro & D'Agostino, 2004*). After calculating the scores and their associated risk, risk groups were designed: low risk (*<5th percentile*), medium risk (from the *5th percentile* to the *median*), high risk (from the *median* to the *95th percentile*), and very high risk ($\geq$ *95th percentile*). All the analyses were done with an $\alpha = 5\%$ and for each relevant parameter the associated confidence interval (CI) was calculated. All the analyses were done with IBM SPSS Statistics 19 and R 2.13.2.

## RESULTS

Of a total of 115 patients who fulfilled the inclusion criteria, three were excluded because there was no further contact after the initial visit (lost during the follow-up). Thus, the final sample comprised 112 patients.

Over a mean follow-up of $2.3 \pm 1.6$ years, 39 of the 112 patients had CVD (34.8%, 95% CI [26.0–43.6%]). Of these, 22 were fatal (19.6%, 95% CI [12.3–27.0%]) (cardiac arrest, 12; ischaemic heart disease, 3; heart failure, 3; stroke, 3; peripheral arterial disease, 1) and 17 were non fatal (15.2%, 95% CI [8.5–21.8%]) (ischaemic heart disease, 7; heart failure, 6; atrial fibrillation, 2; renal failure, 1; pericarditis, 1) (Table 1). This represents an incidence density of 150 CVD for each 1,000 person-years (95% CI [107–206] CVD per 1,000 person-years), of which 104 were fatal (95% CI [69–152] CVD per 1,000 person-years) and 46 non fatal (95% CI [89–273] CVD per 1,000 person-years).

Table 1 shows the descriptive and analytical characteristics of the study patients. The mean age was advanced (70.5 years); the youngest patient was 34 years old. There was a high prevalence of comorbidity (heart failure, 13.4%; renal failure, 8.9%; depression, 8.9%; asthma/COPD, 13.4%; hypertension, 75.0%; dyslipidaemia, 42.9%) and a very high mean number of daily pills (5.6). Concerning lifestyle habits, 21.4% of the patients smoked and 26.8% walked usually. For the diabetes-related variables, 43.8% used insulin and the mean baseline fasting blood glucose was 8.4 mmol/L. Notably, 26.8% of the patients were admitted with a cardiovascular problem.
**Table 1 Baseline characteristics and adjusted hazard ratios for cardiovascular disease for type 2 diabetic inpatients in a Spanish region, 2010–2012 data.**

| Variable | Total (n = 112) n(%)/x ± s | HR | 95% CI | p-value |
|---|---|---|---|---|
| Cardiovascular morbidity: | | | | |
| Ischemic heart disease | 7(6.2) | | | |
| Heart failure | 6(5.4) | | | |
| Atrial fibrillation | 2(1.8) | N/A | N/A | N/A |
| Renal failure | 1(0.9) | | | |
| Pericarditis | 1(0.9) | | | |
| Cardiovascular mortality: | | | | |
| Cardiac arrest | 12(10.7) | | | |
| Ischemic heart disease | 3(2.7) | | | |
| Heart failure | 3(2.7) | N/A | N/A | N/A |
| Cerebral haemorrhage | 3(2.7) | | | |
| Peripheral arterial disease | 1(0.9) | | | |
| Male gender | 59(52.7) | 1.84 | 0.90–3.75 | 0.095 |
| Age (years) | 70.5 ± 12.4 | 1.04 | 1.00–1.08 | 0.031 |
| Depression | 10(8.9) | N/M | N/M | N/M |
| Asthma/COPD | 15(13.4) | N/M | N/M | N/M |
| Hypertension | 84(75.0) | 1.11 | 0.47–2.62 | 0.804 |
| Dyslipidaemia | 48(42.9) | N/M | N/M | N/M |
| Heart failure | 15(13.4) | N/M | N/M | N/M |
| Renal failure | 10(8.9) | 2.76 | 1.01–7.59 | 0.048 |
| Insulin | 49(43.8) | 1.56 | 0.77–3.16 | 0.212 |
| Smoking | 24(21.4) | N/M | N/M | N/M |
| Admission for cardiovascular reasons | 30(26.8) | 2.15 | 1.09–4.25 | 0.027 |
| Pills per day | 5.6 ± 3.9 | N/M | N/M | N/M |
| Habit of walking | 30(26.8) | 0.57 | 0.25–1.31 | 0.185 |
| FBG (mmol/L) | 8.4 ± 4.4 | N/M | N/M | N/M |
| Creatinine (μmol/L) | 97.2 ± 44.2 | N/M | N/M | N/M |

**Notes.**

HR, hazard ratio; CI, confidence interval; COPD, chronic obstructive pulmonary disease; FBG, fasting blood glucose; N/A, not applicable; N/M, not in the model; Goodness-offit of the model: $X^2 = 24.43$, $p < 0.001$; c-statistic, 0.734 (standard error: 0.049).

The HR of the variables included in the stepwise model were: male gender (HR = 1.84, 95% CI [0.90–3.75], $p = 0.095$), older age (per 1 year) (HR = 1.04, 95% CI [1.00–1.08], $p = 0.031$), hypertension (HR = 1.11, 95% CI [0.47–2.62], $p = 0.804$), renal failure (HR = 2.76, 95% CI [1.01–7.59], $p = 0.048$), insulin use (HR = 1.56, 95% CI [0.77–3.16], $p = 0.212$), admission for cardiovascular reasons (HR = 2.15, 95% CI [1.09–4.25], $p = 0.027$) and not having the habit of walking (HR = 0.57, 95% CI [0.25–1.31], $p = 0.185$). The model obtained with these factors was very significant ($p < 0.001$). The scores for each variable in the predictive model and the risk groups are shown in Fig. 1. The c-statistic for the scoring system was 0.734 (standard error: 0.049).

Figure 2 shows that there were significant differences in survival between the various risk groups ($p < 0.001$), with a reduction in survival as the risk category increased.

| Insulin | Score |
|---------|-------|
| Yes | 2 |
| No | 0 |

| Age (years) | Score |
|-------------|-------|
| <40 | 0 |
| 40-49 | 2 |
| 50-59 | 4 |
| 60-69 | 6 |
| 70-79 | 8 |
| ≥80 | 10 |

| Hypertension | Score |
|--------------|-------|
| Yes | 1 |
| No | 0 |

| Habit of walking | Score |
|------------------|-------|
| Yes | -3 |
| No | 0 |

| Gender | Score |
|--------|-------|
| Male | 3 |
| Female | 0 |

| Personal history of renal failure | Score |
|-----------------------------------|-------|
| Yes | 5 |
| No | 0 |

| Admitted due to cardiovascular reasons | Score |
|----------------------------------------|-------|
| Yes | 4 |
| No | 0 |

**Cardiovascular risk (%)**

| Total score | 1 year | 2 years | 3 years | 4 years |
|-------------|--------|---------|---------|---------|
| Low (<5) | <8.00 | <8.35 | <10.85 | <13.85 |
| Medium (5-11) | 9.65-28.95 | 10.10-30.10 | 13.05-37.55 | 16.65-45.75 |
| High (12-18) | 34.10-75.45 | 35.45-77.05 | 43.75-85.55 | 52.65-91.90 |
| Very high (≥19) | >82.00 | >83.45 | >90.55 | >95.35 |

**Figure 1** Four-year risk score for predicting cardiovascular disease in type 2 diabetic inpatients.

# DISCUSSION

## Summary

This study constructed a predictive model for CVD with a good discriminating power (c-statistic = 0.734) indicating which patients with type 2 diabetes who are admitted via the ED have a greater risk of presenting CVD, either fatal or non fatal.

## Strengths and limitations

The main strength of this study is related to the lack of other studies that have constructed short-term predictive models for CVD in patients with type 2 diabetes admitted via the ED. The innovative results can therefore be used to help take decisions to try to avoid the onset of CVD. Additionally, the predictive model constructed had a good discriminating power (c-statistic = 0.734), which will enable precise predictions after validation.

Although the sample size was just 112 patients, it was still sufficient for the aims of this study, as the idea was to evaluate the predictive model and its resulting c-statistic, which indicates the discriminating power of the scale constructed (*Cooney, Dudina & Graham, 2009*). We were therefore very rigorous designing the model, selecting a maximum number of variables with a stepwise procedure. The results obtained in the model indicate a high degree of significance ($p < 0.001$ for the goodness of fit), accompanied by a c-statistic above 70% (0.734). Furthermore, the contrast power in our sample size calculation was 83.28%.

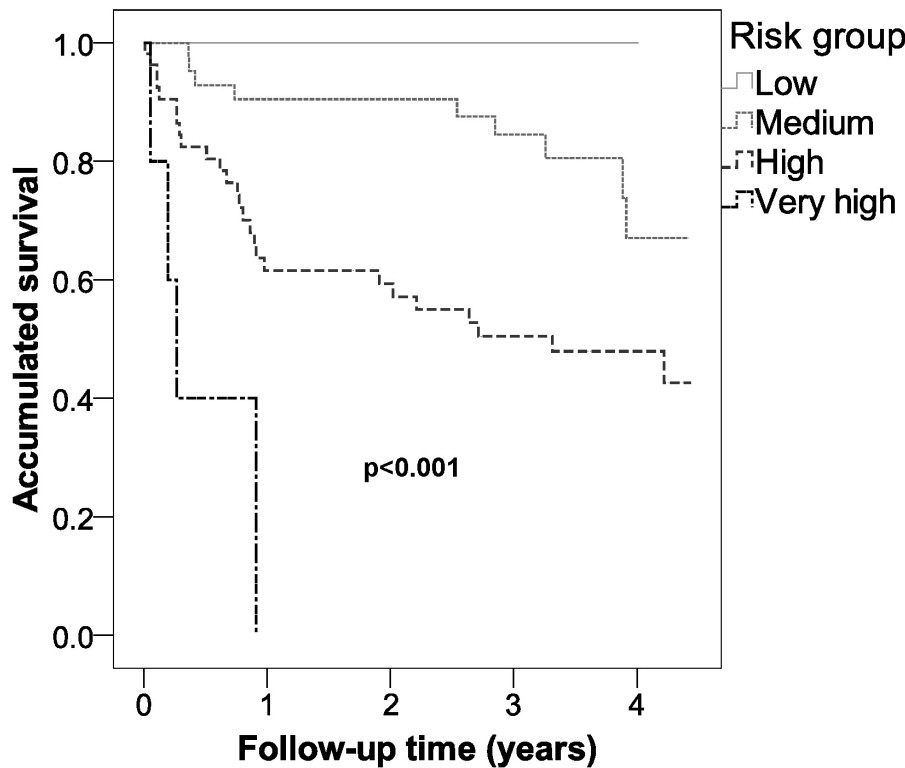

**Figure 2** Survival of the different risk groups for cardiovascular disease of type 2 diabetic inpatients in a Spanish region.

To minimize the possible bias related to measurement and selection, calibrated devices were used and a random sample was selected. However, we were unable to use certain variables that are important in the development of CVD, e.g., obesity, years with diabetes, HbA1c, because the emergency department protocol in our hospital does not include their measurement. If they had been taken into account, then the c-statistic may well have improved. Nevertheless, the resulting value without the inclusion of these variables was satisfactory. Finally, part of the values related to the variable walking habit was obtained by statistical imputation, though the procedure used is considered adequate for this type of model (*Marshall, Altman & Holder, 2010*).

## Comparison with existing literature

Others have constructed cardiovascular risk models that have been extensively validated. However, these models were based on the general population, or patients attending their healthcare centre, working persons or volunteers. Our patients, though, formed a heterogeneous group concerning prognostic factors for CVD among the populations used to construct the existing models. A priori, they were all less stable and all had type 2 diabetes. These differences make comparison with current cardiovascular risk tables very difficult. Nonetheless, the c-statistic for internal validation (0.734) is within the range obtained by the other cardiovascular models (0.708–0.82). This indicates that, if our model

is validated externally with results similar to the internal validation, it could be used in daily clinical practice (*Cooney, Dudina & Graham, 2009*).

The prognostic factors for CVD in our study were: insulin, older age, male gender, renal failure, hypertension, habit of walking, and being admitted for cardiovascular reasons. These results corroborate those of other authors, except for the initial admission due to cardiovascular problems (*Muggeo et al., 2000*; *Bo et al., 2005*; *Hong Kong Diabetes Registry et al., 2008*; *Kleefstra et al., 2008*; *Cooney, Dudina & Graham, 2009*), although this association was very logical. Finally, smoking was notably absent in the predictive model, possibly due to the already high underlying cardiovascular risk of these patients (*Gil-Guillén et al., 2009*).

### Implications for research and/or practice

After validation, this study could provide clinical practice with a tool to predict premature cardiovascular morbidity and mortality in patients with type 2 diabetes admitted via the ED. If our results are confirmed with other studies, those patients who have a high likelihood of CVD within four years should be closely followed with effect from their hospital discharge. The control should be based mainly on medication adjustment, control of therapeutic non-compliance, and ensuring a healthy lifestyle (*Ramírez-Prado et al., 2015*).

This validation will require recruiting a new sample of patients and determining the two key questions with this sample; firstly, whether the scoring system correctly discriminates between those patients who have CVD and those who do not (using the c-statistic), and secondly, whether the proportion of observed events is similar to that given by the model (using $X^2$ tests). This validation is currently the subject of study in our hospital, and obviously it could also be done in other geographical areas, such that if the two previous conditions are verified, a tool will be available to help reduce the incidence of CVD in patients with similar characteristics to those of the present study sample.

## CONCLUSIONS

This study provides a tool that, after validation, will enable short-term cardiovascular morbidity and mortality to be predicted in patients with type 2 diabetes admitted via the ED. This tool should be used by the primary health care services to improve the prognosis, by making more suitable decisions and planning the beneficial needs of the patient, though whenever possible indicating that the patient should walk and carrying out stricter control in those patients who present a high cardiovascular risk.

## ACKNOWLEDGEMENTS

We thank all the services of the General Hospital of Elda who participated in this study. The authors also thank Ian Johnstone for help with the English language version of the text.

### Funding

The authors declare there was no funding for this work.

## Competing Interests

The authors declare there are no competing interests.

## Author Contributions

- Dolores Ramírez-Prado conceived and designed the experiments, performed the experiments, wrote the paper, reviewed drafts of the paper.
- Antonio Palazón-Bru conceived and designed the experiments, performed the experiments, analyzed the data, wrote the paper, prepared figures and/or tables, reviewed drafts of the paper.
- David Manuel Folgado-de la Rosa and Damian Robert James Martínez-St. John conceived and designed the experiments, reviewed drafts of the paper.
- María Ángeles Carbonell-Torregrosa and Ana María Martínez-Díaz conceived and designed the experiments, performed the experiments, contributed reagents/materials/analysis tools, reviewed drafts of the paper.
- Vicente Francisco Gil-Guillén conceived and designed the experiments, contributed reagents/materials/analysis tools, reviewed drafts of the paper.

## Human Ethics

The following information was supplied relating to ethical approvals (i.e., approving body and any reference numbers):

The study posed no additional risk to the patients and an indirect benefit was expected, as the results might reduce short-term cardiovascular morbidity and mortality in this type of patient. The study was carried out in compliance with the principles of the World Medical Association Declaration of Helsinki and complied with the European Union norms of good clinical practice. The patients were informed verbally about the study and about the information required. The study was approved by the Ethics Committee of the Elda Department of Health (Ref. UI13016).

## Supplemental Information

Supplemental information for this article can be found online at http://dx.doi.org/10.7717/peerj.984#supplemental-information.

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
