# Peer review of "A four-year cardiovascular risk score for type 2 diabetic inpatients"

_PeerJ, doi:10.7717/peerj.984_

## Round 0.1 · original submission · Major Revisions

The manuscript is interesting and original, and propose a practical tool to predict the risk for cardiovascular disease in type 2 diabetes. Statistical analysis is, on the whole, acceptable, although the small number of patients enrolled represents a major limitation of this study.
Some methodological concerns should be addressed, that have been outlined by the reviewers. In particular:

Sample size: how was it calculated (statistical software)? Please, discuss.
Why other criteria, including approximate duration of diabetes, and glycemic control (HbA1c) have been not included in your predictive model? Please, discuss.
Why inclusion criteria were age >13? Were the young individuals type 2 diabetics? How was type 1 diabetes excluded? Please discuss.

Reviewer 1 ·

Basic reporting

See comments below.

Experimental design

See comments below.

Validity of the findings

See comments below.

Additional comments

The manuscript of Ramírez-Prado et al. is well written, original and interesting. The laborious and exhaustive statistical analysis is suitable for the study objectives. The authors presented novel results with the construction of a scoring system to predict cardiovascular disease. The figures and the table are self-explanatory and clear.
However, some clarifications are necessary:
1. Introduction:
Line 33: Could you explain better the meaning of "once validated (the model)"? See also line 171.
2. Material & Methods:
Line 64: How did the authors assess the outcome and the renal failure? Please, add references to help readers understand this better.
Lines 65-69: What about the obesity and other parameters, such as years with the diabetes diagnosis? Why were not they included in the predictive model?
Have you reviewed glycated hemoglobin in this study?
4. Sample size and statistical methods:
Lines 83-86: The habit of walking is selected as an important protective variable in the model and it was statistically imputed. Could you mention this in the study limitations?
7. Conclusion:
Could you indicate examples for this phrase: “making more suitable decisions and planning the beneficial needs of the patient”?
8. Tables and figures:
Why did you choose the C-statistics instead of the ROC curve?

Reviewer 2 ·

Basic reporting

The aim of this study is to validate a cardiovascular risk score for diabetic patients arriving to the ER and to propose it as a practical tool for clinician to anticipate mortality risk .
This is an interesting question because most of the cardio-vascular risk score for diabetic or non diabetic patients are long term ( 10 years ) or life long scores.

Experimental design

The experimental design of this study which is a pivotal question of this work and deserves to be discussed by a methodologist ( which I' m not ) seems for me arguable because of the very limted number of patients ( 112) and events ( 39) .

The presentation of the paper is acceptable

Validity of the findings

Very limited without replication or complementary studies

Additional comments

The authors should provide a more precise justification of on the methological basis of their interesting work and how such a limited number of diabetic patients could validate such a model of cardio-vascular risk score

---

## Round 0.2 · accepted · Accept

The authors have fulfilled to all the revision's requests.

Reviewer 1 ·

Basic reporting

No comments.

Experimental design

No comments.

Validity of the findings

No comments.

Additional comments

The authors have addressed correctly all the questions, resulting in an interesting and relevant paper, which has improved compared to the first version.